Antioxidant enzyme cycling over reproductive lunar cycles in Pocillopora damicornis

Murphy James W.A. jwmurphy@hawaii.edu 1 2
Collier Abby C. 3
Richmond Robert H. 1
1 Kewalo Marine Laboratory, Pacific Biosciences Research Center, University of Hawaii at Manoa , Honolulu , HI , United States of America
2 Department of Molecular Biosciences and Bioengineering, University of Hawaii at Manoa , Honolulu , Hawaii (HI) , United States of America
3 Faculty of Pharmaceutical Sciences, University of British Columbia , Vancouver , British Columbia , Canada
Kelly Linda
Electronic publication date: 2019 Jun 7
Publication date: 2019
Volume: 7
Electronic Location ID: e7020
Received 2018 Mar 21; Accepted 2019 Apr 25
Copyright: ©2019 Murphy et al.
Copyright year: 2019
Copyright holder: Murphy et al.
License: This is an open access article distributed under the terms of the Creative Commons Attribution License, which permits unrestricted use, distribution, reproduction and adaptation in any medium and for any purpose provided that it is properly attributed. For attribution, the original author(s), title, publication source (PeerJ) and either DOI or URL of the article must be cited.
License URL: https://creativecommons.org/licenses/by/4.0/

Keywords: Enzymes, Glutathione peroxidase, Superoxide dismutase, Lunar cycling, Coral, Glutathione reductase, Antioxidant defense, Catalase

Funding: National Oceanic and Atmospheric Administration NA09NOS4780178 Edmondson Research Fund (University of Hawai‘i at Mānoa Department of Biology) National Fish and Wildlife Foundation 2008-0061-016 This project was financially supported through National Oceanic and Atmospheric Administration (grant number NA09NOS4780178), the Charles H and Margaret B. Edmondson Research Fund (University of Hawai‘i at Mānoa Department of Biology), and National Fish and Wildlife Foundation (grant number 2008-0061-016). The funders had no role in study design, data collection and analysis, decision to publish, or preparation of the manuscript.

==============================
The impacts of continued degradation of watersheds on coastal coral reefs world-wide is alarming, and action addressing anthropogenic stressors and subsequent rehabilitation of watersheds and adjacent reefs is an urgent priority. The aim of this study is to develop and improve the use of antioxidant enzymes as bioindicators of stress in coral species. In order to fully develop such tools, it is necessary to first understand baseline cycling of these enzymes within coral tissues. Due to inherent links between reproduction and oxidative stress, these aims may be facilitated by sampling coral tissues over reproductively-linked lunar cycles to determine variations from baseline. By developing a greater understanding of biochemical markers of stress in corals, specifically antioxidant defense enzymes catalase (CAT), glutathione reductase (GR), glutathione peroxidase (GPx), and superoxide dismutase (SOD) in Hawaiian Pocillopora damicornis, we have provided molecular tools that identify thresholds of stress on coral reefs. Our results suggest that the coral reproductive state is a significant factor affecting the activity of antioxidant enzymes. Specifically, CAT and GR display maximum activity during peak reproductive state. Whereas significant maximal Se-independent GPx and SOD activity was measured during off-peak reproductive cycles. Such insight into the cyclical variation of the activity of these enzymes should be applied towards differentiating the influence of natural biological activity cycling in diagnostic tests identifying the effects of different physical environmental factors and chemical pollutants on coral health. Through the development and application of these molecular biomarkers of stress, we look to improve our ability to identify problems at the sub-lethal level, when action can be taken to mitigate a/biotic impacts.

Introduction

Corals are critical to the structural and biological integrity and function of coral reef ecosystems (Birkeland, 1997). Because anthropogenic stress is increasingly impacting global marine environmental health (Gattuso et al., 2015; Heron et al., 2016; Hughes et al., 2017; Maynard et al., 2015), it is vital that techniques for evaluating coral stress (prior to reef collapse) are developed and applied (Edge et al., 2013). Recent advances in the application of molecular analyses to facilitate sub-lethal stress evaluations in corals have been substantial (Ainsworth et al., 2008; Barshis et al., 2014; Desalvo et al., 2008; Downs et al., 2012; Edge et al., 2013; Rougée et al., 2006). Researchers concerned with coral health, stress, bleaching events and their total effects on the state of reef health, need to develop new and better diagnostic tools that address coral stress prior to death and that can inform policy, improve conservation efforts, and assist in saving coral reefs as a legacy for the future.

One way that this can be accomplished is by developing tests for the evaluation of specific stress responses. Antioxidant stress enzymes, for example, are useful for the analysis of stressor impacts on the health of coral animals due to their involvement in physiological responses to a variety of stressors (Downs et al., 2006; Higuchi, Yuyama & Nakamura, 2015; Vijayavel et al., 2012). Antioxidant enzyme presence and activity in coral tissues have the potential to be employed as metrics for evaluating stress on reefs, including gradients of stress, and pin-pointing the impacts of toxicants on the health of corals (Edge et al., 2013; Rivest & Hofmann, 2014). Such molecular biomarkers can determine the degree of stress affecting specific areas of the reef or the gradient over which a pollutant source may be diffusing across a reef (Downs et al., 2006). A library of biomarkers will be valuable in identifying physiological stress prior to coral death.

Previous studies have highlighted antioxidant enzymes as useful biomarkers of the impacts of stressors such as heat, xenobiotic exposure, and high-irradiance (Downs et al., 2006; Higuchi, Yuyama & Nakamura, 2015; Liñán Cabello et al., 2010a; Liñán Cabello et al., 2010b; Olsen et al., 2013). However, as useful as this suite of enzymes is in providing information about coral stress responses and threat levels, many of the substrates that trigger this type of stress are naturally produced in normal homeostatic processes (Agarwal, Gupta & Sikka, 2006; Dowling & Simmons, 2009; Fujii, Iuchi & Okada, 2005). Adding these biomarkers to the available tools that can be employed to evaluate reef health is important and requires knowledge of baseline levels of protein expression and activity, including measurements over reproductive cycles. In order to have full confidence in using these enzymes as biomarkers for stress detection, it is important to take into consideration how endogenous levels of activity may change over shifting baselines. As such, prior to expanding our use of these enzymes as diagnostic tools for oxidative stress responses, we seek to characterize whether reproductive cycling has a discernable effect on the enzymatic profile of a widely distributed species of coral, Pocillopora damicornis (Linnaeus).

This study of coral reproduction in relation to baseline shifts in enzyme activity arose in part from publications describing cyclical variation in the activity of xenobiotic metabolizing enzymes during reproduction events. A study performed by Rougée, Richmond & Collier (2014) illustrated variations in the expression and activity of xenobiotic metabolizing enzymes during reproductive cycling in the coral P. damicornis. Glucuronosyltransferase, glutathione-s-transferase (GST), cytochrome P450 2E1, and cytochrome P450 reductase were all found to fluctuate significantly over natural reproductive lunar cycles (Rougée, Richmond & Collier, 2014). Additionally, research by Ramos et al. (2011) provided insight into the effect of reproductive cycling on various biotransformation and antioxidant enzyme activities. In their work, activities of cytochromes P450, GST, NADPH c reductase, and catalase (CAT) were all significantly higher during reproductive peaks in the coral Siderastrea siderea (Ramos et al., 2011). With such evidence for the fluctuation of enzymatic activity tied to reproductive cycling, coupled with the knowledge of reactive oxygen species (ROS) impacts on the health of reproductive systems in non-cnidarian species, the lack of more comprehensive research into antioxidant enzyme expression over reproductive cycles in corals underlines a lack of data regarding antioxidant enzyme expression (Agarwal, Gupta & Sikka, 2006).

Reproduction is an innate source of ROS generation and relies heavily upon the interplay of pro-oxidants and antioxidants (Agarwal, Gupta & Sikka, 2006; Fujii, Iuchi & Okada, 2005; Halliwell & Gutteridge, 2015; Rahal et al., 2014). This interplay of ROS production and detoxification during reproduction has a critical role in both aiding and inhibiting high quality gamete production, fertilization, and embryo development (Fujii, Iuchi & Okada, 2005). Studies in non-Cnidarians have pointed to a heightened prevalence of ROS impacting fertility, as well as being implicated in the termination of embryos and reproductive senescence during heightened levels of oxidative stress (Agarwal, Gupta & Sharma, 2005; Agarwal, Gupta & Sikka, 2006; Carbone et al., 2003). Oxidative stress also has the potential to reduce embryo growth and decrease fertilization rates (Agarwal, Gupta & Sikka, 2006). However, ROS are also both beneficial and detrimental to the motility and viability of sperm cells. Specifically, sulfoxidation is required for the maturation of sperm and packaging of nuclei in sperm heads, while excess ROS proliferation acting upon the axoneme of spermatozoa can inhibit motility (De Lamirande & Gagnon, 1992; Fujii, Iuchi & Okada, 2005). Rich in polyunsaturated fatty acids, spermatozoa are highly vulnerable to lipid peroxidation due to low availability of ROS-scavenging enzymes (Agarwal, Gupta & Sikka, 2006; Saleh & Agarwal, 2002). As a result, unregulated lipid peroxidation can lead to the production of spermicidal compounds, such as (E)-4-Hydroxy-2-nonenal, which at concentrations of only 50 µm, can result in irreversible motility loss (Selley et al., 1991). Antioxidant compounds, such as glutathione, and ROS-scavenging enzymes, such as superoxide dismutase (SOD), aid in modulating the effects of ROS on egg and sperm viability and promote embryo integrity (Agarwal, Gupta & Sikka, 2006). Although corals may have different reproductive methods than vertebrates, other invertebrates, and plants, there are commonalities with respect to ROS generation and detoxification that are highly conserved across taxa and are required for optimizing reproductive integrity (Dowling & Simmons, 2009). To improve the breadth and quality of the biomarkers available, this project sought to define basal enzymatic activity levels in a major coral species with broad global distribution, with respect to reproductive cycling (Hoeksema, Rodgers & Quibilan, 2014).

Consistent with the aim of this study to characterize antioxidant enzyme activity over reproductive cycling in corals, colonies of Hawaiian P. damicornis (type-B) were chosen for study due to their documented monthly brooding cycles exhibiting peak planula output closely tied to the first-quarter moon phase (Kolinski & Cox, 2003; Richmond & Jokiel, 1984; Schmidt-Roach et al., 2012; Stimson, 1978). Characteristic monthly reproductive cycling of P. damicornis in Hawaii and the Pacific Islands, suggests this coral as an optimal candidate for study in comparison to other common reef-building corals, such as Porites spp., Acropora spp., and Montipora spp., that seasonally spawn over annual cycles (Harrison et al., 1984; Harrison & Wallace, 1990; Neves, 2000; Padilla-Gamiño & Gates, 2012; Stimson, 1978). As such, potential reproductive shifts in antioxidant enzyme activity may be observed over monthly cycles, rather than over an annual reproductive cycle. This also makes the species an excellent candidate for differentiating seasonal variations and year-to-year changes in basal antioxidant enzyme activity variation (Cooper, Gilmour & Fabricius, 2009; Harrison & Wallace, 1990; Ward, 1995). By improving baseline knowledge of endogenous cycles in cellular physiology of major reef-building corals through testing the hypothesis that reproductively-tied lunar cycling has an effect on antioxidant enzyme activity, future studies examining coral health can better identify stress endured by corals with respect to extreme environmental and anthropogenically-influenced phenomena from natural metabolic enzyme activity cycling.

Methods

Sample collection

Coral samples (5 cm × 2.5 cm nubbins) were collected periodically from the same colonies under Department of Land and Natural Resources—Division of Aquatic Resources coral collection permit SAP 2015-6/17 off Lilipuna Pier, Kāne‘ohe, O‘ahu, Hawai‘i (Fig. 1), adjacent to the Hawai‘i Institute of Marine Biology (HIMB).

Figure 1 Site map denoting locations of the 6 Pocillopora damicornis colonies of interest in this study.

Sampled colonies were distributed over a 60 m transect in southern Kāne‘ohe Bay, O‘ahu, Hawai‘i. Photo credit: 2018 Google Earth.

To reduce the impact of fragmentation on the reproductive cycling or output of P. damicornis (Zakai, Levy & Chadwick-Furman, 2000), colonies were not fragmented prior to the start of collections. Instead, fragments of branches were sampled from several areas on each colony and included tissue extending from central to distal areas of each branch to ensure reduction of microhabitat influence and intracolony stress load variation between samples. This was also done to limit variations in reproduction potential along coral branches, as polyps found mid-branch retain the highest planula larvae output versus distal and central branch polyps (Harrison & Wallace, 1990). Sampling was also conducted during falling tides to both reduce residence time in low-flow water and match peak planula release, as it has been correlated with low tide periods (Harrison & Wallace, 1990). Colonies with minimal competition from other corals and no visible signs of disease or stress were sampled during each moon phase (New, 14, Full, 34 moon; n = 6 colonies) during July and August. Collections also included an acute sampling period, during which corals were sampled daily for five consecutive days following the start of the peak reproductive period moon phase (14 moon) in the month of August. This sampling period was constructed to provide finer resolution for understanding changes in antioxidant enzyme profiles following a reproductive peak.

Fragments were immediately frozen in liquid nitrogen, transported to Kewalo Marine Laboratory (KML) on dry ice, and transferred to a −80 °C freezer to preserve enzyme profiles and protein integrity. Prior to protein extraction, corals were crushed into a fine powder using liquid nitrogen and an arbor press.

S9 protein fraction extraction and protein quantification

Following modified protocols by Lesser et al. (1990), coral S9 post-mitochondrial protein fractions were isolated from crushed coral tissue. Using 1,500 mg of crushed tissue and 1,500 µL of homogenization buffer per sample extraction in 50 mL tubes (0.01 M Tris–HCl buffer pH 8.0, 1 M phenylmethylsulfonyl fluoride in 1% v/v dimethyl sulfoxide), tissue was homogenized for 1 min on ice using an Ultra-Turrax homogenizer. The homogenate was then spun for 5 min at 4 °C at 2,000 rcf in an Eppendorf Microcentrifuge 5415D (Hauppauge, NY, USA) to separate skeleton and tissue, and the supernatant was transferred to 1.5 mL microcentrifuge tubes and spun for 20 min at 4 °C at 10,000 rcf. The supernatant was then aliquoted into 1.5 mL tubes and frozen at −80 °C; 50 µL of each extracted sample was set aside for protein concentration analyses.

In preparation for enzymatic activity assays, protein concentrations from each sample were measured using a bicinchoninic acid (BCA) assay. The standard curve was constructed with bovine serum albumin from 0 to 1.0 mg/mL protein (25 µL/well in triplicate), 1:5 dilutions of aliquots from each extracted S9 sample fraction in double distilled water (ddH2O) were loaded into a 96 well plate in triplicate (25 µL/well). Bicinchoninic acid development reagent (2% Cu2+SO4 in BCA; Sigma-Aldrich) was then added into each well (200 µL/well), and the loaded plate was incubated at 37 °C for 30 min. Following incubation, absorbance values were determined at λ = 562 nm in a SpectraMax M5 Micro-Plate spectrophotometer (Molecular Devices, Sunnyvale, CA, USA). To ensure triplicate absorbance variation was within acceptable experimental levels (percent coefficient of variation, %CV <10%; coefficient of determination, R2 > 0.98), data were then exported to Microsoft Excel and %CV and subsequent standard curve R2 values were calculated. Sample protein concentration values were then interpolated from the standard curve; those extractions falling below 1 mg/mL required re-extraction of S9 post-mitochondrial protein fractions.

Enzyme activity assays

Enzyme assays were developed in-house through modifications of protocols for CAT, GR, GPx, and SOD (Aebi, 1984; Lawrence & Burk, 1976; McCord & Fridovich, 1969; Regoli, Bocchetti & Filho, 2012), and chemicals for assays were sourced from Sigma-Aldrich (St Louis, MO, USA), BioVision (Zurich, Switzerland), EMD Millipore (Burlington, MA, USA), and Cayman Chemical (Ann Arbor, MI, USA). Assays were analyzed using a SpectraMax M5 Multi-Plate reader and final activity values calculated using SoftMax Pro and Microsoft Excel.

The metabolism of H2O2 as a marker for CAT activity was accomplished by measuring the consumption of H2O2 over time by analyzing decreasing absorbance of H2O2 at λ = 240 nm. Briefly, the method was performed as follows: on ice, coral protein extractions were first diluted to 1 mg/mL in 50 mM potassium phosphate buffer pH 7.0 and then loaded, in triplicate (10 µL/well), into optically clear microtiter 96-well plates; negative controls were also run in triplicate containing all assay reagents except S9 protein to correct for spontaneous H2O2 degradation during activity reads. Running buffer (50 mM potassium phosphate buffer pH 7.0) was then loaded into wells (80 µL/well), and samples were incubated for 3 min at 25 °C. The CAT activity reaction was then initiated by adding 10 µL of 120 mM H2O2 to each well, and immediately transferring the reaction plate into the spectrophotometer to read at 10 s intervals for 5 min. In order to dislodge O2 bubbles created by the dismutation of H2O2 into H2O and O2, the spectrophotometer was set to vibrate the 96-well plate between 10 s reads for 2 s; this aided in preventing O2 bubbles from obscuring the plate reader’s evaluation of H2O2 absorbance in the reaction wells.

To evaluate the activity of GR, the consumption of NADPH at λ = 340 nm was observed over time as GR in coral samples consumed this co-factor during the reduction of the reagent oxidized glutathione (GSSG). In order to account for both spontaneous degradation of NADPH in reaction wells and endogenous concentrations of NADPH in coral samples, wells containing no coral sample (spontaneous degradation control), and those with coral sample but no NADPH (background level control), were also evaluated alongside wells containing all reagents; in place of coral sample and NADPH, an extra 20 µL of 100 mM potassium phosphate buffer (pH 7.2) was added to wells in the first assay step. Values for NADPH degradation obtained from these controls were subtracted from overall activity following assay completion. In optically clear microtiter 96-well plates, 100 mM potassium phosphate buffer was loaded into wells (130 µL/well, in triplicate), followed by 100 mM ethylenediaminetetraacetic acid (EDTA) in ddH2O (10 µL/well), 10 mM GSSG in ddH2O (20 µL/well), 1.2 mM NADPH in ddH2O (20 µL/well), and 1 mg/mL coral sample (20 µL/well). Plates were then loaded into the spectrophotometer and mixed using its mixing function for 5 s. Absorbance reads were conducted at 20 s intervals for 5 min at 25 °C.

To evaluate the activity of GPx, assays were broken into two parts in order to determine the activity of both selenium-dependent and selenium-independent forms of this enzyme. As such, the loading protocol in place for evaluating the activity of selenium-dependent GPx employed H2O2 as the initiator and substrate for this reaction, including sodium azide (NaN3) to inhibit CAT activity from consuming H2O2 and interfering with assay results. To evaluate the activity of selenium-independent GPx, cumene hydroperoxide (CHP) was utilized in place of H2O2; NaN3 was not employed as a CAT inhibitor due to the change in substrate, and buffer volumes were adjusted to bring the final reaction volume to 200 µL. In order to achieve the breakdown of their substrates, GPx use reduced glutathione (GSH) as the cofactor for hydroperoxide reduction, producing oxidized glutathione as the final product (GSSG). In order to visualize this breakdown and quantify GPx activity, this assay has been adapted to measure the consumption of reduced NADPH by GR to replenish GSH from the GPx by-product, GSSG. By this method, measured decreases in NADPH are proportional to GPx activity, which is monitored at λ = 340 nm for 5 min at 20 s intervals in optically clear 96-well microtiter plates. Three sets of reference wells were run to account for: degradation of assay substrates over time (no coral sample with substrates), non-specific oxidation of NADPH in this assay (no H2O2 or CHP), and endogenous levels of substrate in coral tissue samples (coral sample with no substrate). Assay reagents consisted of: 100 mM potassium phosphate buffer (pH 7.0; 130, 140, 110, and 120 µL/well for reference, no H2O2, coral sample, and blank wells, respectively), 20 mM NaN3 working solution in ddH2O (10 µL/well; only for Se-dependent GPx assays), 100 mM EDTA in ddH2O (10 µL/well), 100 mM GSH working solution in ddH2O (20 µL/well), 100 U/mL GR working solution in ddH2O (10 µL/well), 1 mg/mL coral samples in 100 mM potassium phosphate buffer (pH 7.0; 20 µL/well in coral sample and blank wells only), 1.25 mg/mL NADPH working solution in ddH2O (10 µL/well), 150 mM H2O2 working solution (10 µL/well in reference and coral sample wells only), and 150 mM CHP working solution (10 µL/well in reference and coral sample wells only; only for Se-independent GPx assays). Upon loading plates into the spectrophotometer, plates were mixed for 5 s, and absorbance changes immediately read.

Since SOD catalyzes the dismutation of O2− to H2O2 and O2, this assay conducted an indirect evaluation of SOD activity by analyzing the degree of inhibition of the reduction of cytochrome c by O2−. Reagents and samples were loaded into a 96-well microtiter plate in triplicate, loaded into and mixed using the mixing function in the spectrophotometer for 5 s, and analyzed for decreasing absorbance (λ = 550 nm) for 5 min in 20 s intervals at 25 °C. For this assay, wells were loaded in the following order: 100 mM potassium phosphate buffer (pH 7.8, 80, 74, 94 µL/well for reference, coral sample, and coral blank wells, respectively), working buffer (100 mM potassium phosphate buffer (pH 7.8), 0.2 mM EDTA in ddH2O, 100 µM hypoxanthine in ddH2O, and 20 µM cytochrome c in ddH2O; 100 µL/well), coral samples diluted to 1 mg/mL in 100 mM potassium phosphate buffer (pH 7.8; 6 µL/well in coral sample and blank wells only), and reactions were initiated with the addition of 300 mU/mL of O2− generating xanthine oxidase in ddH2O (20 µL/well in reference and coral sample wells only).

Statistical analyses

Statistical data analyses were conducted using Prism 7.03 (GraphPad Software, La Jolla, CA, USA). Data normality was evaluated using the D’Agostino and Pearson Omnibus normality test. Data were then run through one-way analyses of variance (ANOVA) with Tukey’s post-hoc test to elucidate significant variances between sample means; p < 0.05, α = 0.05, CI = 95%. Under normal circumstances, repeated measures tests would be suitable for such statistical analysis due to sampling of the same colonies over time. However, several issues with volatility of ROS and products during kinetic assays led to the exclusion of samples where triplicate runs experienced abnormal jumps in enzyme activity. As such, repeated measures tests could not be accomplished, and one-way ANOVAs were instead conducted. Waveform analyses were run to confirm sinusoidal cycling of enzyme activity with respect to moon phase and determine peaks and troughs in enzyme activity under reproductive cycling. Such analyses were conducted utilizing a sine wave analysis fitting data using least squares regression. These were complemented by electing symmetrical, asymptotic confidence intervals (CI = 95%) and data goodness-of-fit was visualized through R squared values. These data elucidate biological minima and maxima for enzyme activity, where data below or above such values illustrate suppression or induction of enzyme activity by external factors (“perfect fit” of waveform models was determined by frequency = 1). Residual plots were constructed to confirm error was random.

Results

Catalase enzyme kinetic assay analyses displayed a notable trend on enzyme activity cycling over moon phase cycles (Fig. 2A). Although significant variations in enzyme activity were only observed between July Full moon and August 14 moon sampling periods (p = 0.0044, CI = 95%), CAT activity follows a general sinusoidal trend, with activity peaking during the new and 14 moon phases. Resultant information from waveform analyses suggest that baseline reproductive CAT activity is 52.94 ± 0.23 mmol/min/mg protein (amplitude = 12.98 ± 0.33 mmol/min/mg protein, frequency = 1.117). Peak biological CAT activity is 65.92 mmol/min/mg protein, while minimum calculated biological CAT activity fitting this model is 3.99 mmol/min/mg protein. No significant variations in enzyme activity were observed during the acute sampling timeline (p = 0.2374, CI = 95%; Fig. 3A). This was expected, however, as there was no significant difference between August 14 and full moon collections.

Figure 2 Antioxidant enzyme activity versus moon phase cycle.

(A) CAT activity characterized by the consumption of H2O2 mmol/min/mg protein, (B) GR activity characterized by the consumption of NADPH nmol/min/mg protein, (C) Se-independent GPx activity characterized by the consumption of CHP nmol/min/mg protein, (D) Se-dependent GPx activity characterized by the consumption of H2O2 nmol/min/mg protein, and (E) SOD activity characterized by the inhibition of cytochrome c reduction mmol/min/mg protein versus moon phase cycle (planulation peak during 14 moon phases) (error bars represent mean ± SD, treatments that share the same letter are not significantly different p = 0.05).

Figure 3 Antioxidant enzyme activity versus time after peak reproduction (14 moon phase).

(A) CAT activity characterized by the consumption of H2O2 mmol/min/mg protein, (B) GR activity characterized by the consumption of NADPH nmol/min/mg protein, (C) Se-independent GPx activity characterized by the consumption of CHP nmol/min/mg protein, (D) Se-dependent GPx activity characterized by the consumption of H2O2 nmol/min/mg protein, and (E) SOD activity characterized by the inhibition of cytochrome c reduction mmol/min/mg protein versus time following peak reproduction (error bars represent mean ± SD, treatments that share the same letter are not significantly different p = 0.05).

Glutathione reductase activity reflected similar trends as CAT assays, where new and 14 moon phases harbored higher enzyme activity than comparative full and 34 moon phases (Fig. 2B). Activity of GR was significantly higher during July new and 14 moon and August 14 moon than those values from full and 34 moon collection periods (p < 0.0001, CI = 95%). Waveform analyses defined baseline biological GR activity as 9.55 ± 1.02 nmol/min/mg protein (amplitude = 3.09 ± 1.37 nmol/min/mg protein). Maximum and minimum biological GR activity were defined as 12.64 and 6.46 nmol/min/mg protein, respectively (frequency = 4.83). Glutathione reductase activity was found to significantly decrease from day 1 to day 5 of acute sampling following the August 14 moon (p = 0.0189, CI = 95%, Fig. 3B).

Se-independent GPx activity was found to have significant peaks in activity during both July and August 34, and August full moon phases (p = 0.0001, CI = 95%, Fig. 2C). Inverse to the trends of CAT and GR activity with relation to moon phase cycle, GPx was found to follow a sinusoidal activity curve, with peak activity occurring opposite to P. damicornis peak reproductive output. Waveform analysis defined baseline Se-independent GPx activity as 3.28 ± 0.57 nmol/min/mg protein (amplitude 2.07 ± 0.82 nmol/min/mg protein). Maximum and minimum biological Se-independent GPx activities were 5.35 and 1.21 nmol/min/mg protein (frequency = 1.72). When analyzing values for acute variations in GPx activity following the August 14 moon, activity significantly increased in the days following the 14 moon phase (Fig. 3C). However, only activity of August 14 moon day 3 samples were found to be significantly different than other collection time points, having significantly higher activity than those collected on day 1 of the August 14 moon (p = 0.031, CI = 95%). Interestingly, Se-dependent GPx activity was negligible or not detectable across all moon phases and throughout the acute collections (p > 0.05, CI = 95%, Figs. 2D and 3D, respectively). Selenium-dependent GPx did not fit waveform analyses due to multiple zeroes for enzyme activity data.

Values for SOD activity, which are inversely proportional to the degree of cytochrome c oxidation over time, showed significantly higher activity during the July full moon cycle versus all other sampling time points (p = 0.0454–0.0002, CI = 95%, Fig. 2E). As such, SOD activity is demonstrated as being highest following the reproductive peak of P. damicornis. Data showed good fit for waveform analysis (frequency = 0.91), defining baseline biological SOD activity as 139.50 ± 6.23 mmol/min/mg protein (amplitude = 15.42 ± 8.47 mmol/min/mg protein). Maximum and minimum biological SOD activity is 154.92 and 124.08 mmol/min/mg protein. Acute sampling analysis displayed day 5 of the August 14 moon as having significantly higher SOD activity versus days 3 and 4 (p = 0.0351 and p = 0.0004, respectively, CI = 95%, Fig. 3E). However, day 5 SOD activity was not significantly different than measured activity from day 1 and day 2 (p > 0.05, CI = 95%). What is more, day 2 activity was significantly higher than that calculated during day 4 (p = 0.0224, CI = 95%), suggesting that SOD activity dropped significantly before increasing again between moon phases.

Discussion

In order to develop rapid and efficient tools for detecting sub-lethal levels of stress in corals, first there is a need to define foundational changes in coral protein expression and activity patterns across normal homeostatic processes (Downs et al., 2012; Rougée, Richmond & Collier, 2014). Without such definition, there is potential to mistake significant variations in coral health for responses to stress exposure, rather than those due in-part to normal biological processes based on the timing of coral sampling with respect to baseline (Rougée et al., 2006). For example, there would be great importance and versatility in the use of CAT, GR, SOD, and GPx for defining ROS-induced stress in coral animals, which aids in better defining such information as seasonal stress variations, xenobiotic impacts, and thermal stress limitations in corals (Downs & Downs, 2007; Griffin, Bhagooli & Weil, 2006; Higuchi et al., 2008; Lesser, 1996; Liñán Cabello et al., 2010a; Liñán Cabello et al., 2010b). Work to characterize baseline homeostatic levels of activity of these enzymes will benefit the coral conservation biology community and aid in improving experimental design by accounting for changes in the background activity levels of antioxidant enzymes due to innate biological processes. Further, results from this study help bolster this effort to assess fluctuations in baseline antioxidant enzyme activity levels, as this study found activity values significantly varied in relation to reproductive cycling for CAT, GR, SOD, and GPx. Effectively, this study contributes to reproductive biology as well as conservation and toxicology.

Similarly to the findings of Ramos et al. (2011) in S. siderea, a significant peak in CAT activity was observed with relation to peak reproduction in P. damicornis (Fig. 2A). Though acute sampling did not detect day-to-day changes in activity values following peak planulation in August (Fig. 3A), August 14 moon (peak reproductive output) CAT activity was significantly higher than that of July full moon (off-peak reproductive output) activity values (p = 0.0177, CI = 95%). Waveform analysis data points to well-defined cyclical activity based on the lunar phase, with maximum biological CAT activity occurring during the 14 moon cycle. The implications of these findings for CAT activity are in accordance to what has been observed in other coral species and organisms: reproduction is a process during which endogenously generated ROS are produced in P. damicornis. Further, it must be considered that there exist biological minima and maxima for CAT activity (39.96–65.92 mmol/min/mg protein) relevant to the understanding of induction or suppression of this enzyme over a cyclical reproductive scale. Subsequent investigations utilizing CAT activity as a marker for oxidative stress should consider minima and maxima for activity respective to sampling phase.

The finding that significant increases in GR activity are associated with reproductive peaks (p = 0.0001, CI = 95%, Fig. 2B) provides further evidence for the need to consider reproductive time points when using antioxidant enzymes as biomarkers for oxidative stress evaluation. In conjunction with these findings, results illustrating significant decreases in GR activity over the 5-day acute sampling period provide better resolution in identifying the rate at which antioxidant enzyme activities can significantly change over natural P. damicornis brooding cycles with activity during day 5 significantly lower than day 1 (p = 0.0162, CI = 95%, Fig. 3B). Results of GR activity assays provide useful insight into the replenishment of the powerful antioxidant, reduced-glutathione, under reproductive pressures. Similarly to CAT activity measurements, peak and trough data for enzyme activity help demonstrate general trends in naturally cycling GR activity levels in this coral species. These findings suggest that under reproductive peaks, enzymes are utilizing glutathione to reduce ROS to less reactive forms at a more rapid pace. Such evidence would propose that enzymes, such as GPx would display peaks in activity during reproduction, accordingly. However, this was not observed in sampled colonies.

The enzyme Se-independent GPx displayed significantly greater activity during full and 34 moon, rather than new and 14 moon phases (Fig. 2C). Findings for GPx activity maxima and minima during reproductive peaks and troughs will guide future study design employing Se-independent GPx in P. damicornis stress response detection. However, unlike Se-independent GPx, Se-dependent GPx was not significantly active, as activity assays found little to no detectable activity during the full study period (Figs. 2D and 3D). It is possible that associated ROS production during reproductive peaks is not a strong enough driver to elicit synthesis of the Se-dependent form of GPx, or that enzymes, such as CAT, are favored as primary responders to low-levels of ROS within coral tissues. These findings are confounding, as activity patterns inversely mirror those expressed in both CAT and GR. Further, it would be expected that significant increases in GR activity would be proportional to those of GPx enzymes, as the two enzymes work in apposition. However, similar findings illustrating off-peak maxima in the activity of antioxidant enzymes have been documented through the work of Rougée, Richmond & Collier (2014), wherein glutathione-s-transferase, which also utilizes reduced glutathione for pro-oxidant detoxification, and the UDP-glucuronosyltransferase family expressed significantly higher activity 2 weeks following planulation in P. damicornis. Additionally, while GPx-1 is a widely expressed GPx isozyme commonly located within cellular cytosol, mitochondria, and nucleus, it is possible that corals employ different forms of this enzyme for cellular detoxification (Margis et al., 2008). Such variation in the utilization of alternate isozymes has been described in studies of other marine invertebrates, including corals, with respect to their response to hypoxia and anaerobic respiration (Eberlee, Storey & Storey, 1983; Fields, 1983; Fields et al., 1980; Murphy & Richmond, 2016; Plaxton & Storey, 1982).

Building upon trends observed for GPx activity, SOD activity was found to experience a small, but significant increase during the July full moon phase versus all other sampling periods with the highest measured inhibition of cytochrome c reduction (p = 0.0005, CI = 95%, Fig. 2E). Additionally, acute sampling following the August 14 moon also found activity to significantly vary, as days 3 and 4 expressed significantly lower activity versus day 5 (p = 0.0009, CI = 95%, Fig. 3E). This variation highlights potential acute day-to-day shifts in SOD activity unrelated to reproductive cycling. Supplementary investigations of daily shifts in SOD activity over monthly cycles would help clarifying if detected significant decreases in SOD activity are attributable to variations in environmental conditions during sampling or are primarily driven by reproductive peaks. Waveform analyses aided in defining strong cyclical trends in SOD activity with relation to reproductive troughs. Well-defined maximum and minimum values for SOD activity instruct future analysis focused in defining significant changes in coral health in response to external pressures. Additionally, relatively low amplitude in SOD activity cyclicity versus CAT activity suggests that the activity of SOD is more tightly regulated, such that this enzyme may serve a more specialized function during high ROS load versus that of CAT, which may have more diverse biological applications in ROS detoxification. However, like the GPx family of enzymes, SOD harbors a variety of isozymes that contribute to ROS detoxification and may exhibit characteristically different activity during the reproductive cycle leading to these findings. The design of this study sought to isolate Cu/Zn SOD activity through the exclusion of zooxanthellate and mitochondrial tissue fractions, as this isozyme is generally localized within the cytosol of cells (Asada, Kanematsu & Uchida, 1977; Fukai & Ushio-Fukai, 2011). Though Fe and Mn SOD isozymes are generally localized within chloroplasts and mitochondria, respectively, Richier et al. (2003) has described the presence of Fe and Mn SODs within endodermal cnidarian tissues in the anemone Anemonia viridis and Stylophora pistillata (Asada, Kanematsu & Uchida, 1977; Fukai & Ushio-Fukai, 2011). This suggests finer scale improvements should be implemented in future SOD enzyme kinetic assays to isolate activity differences between Cu/Zn, Fe, and Mn SOD from these tissue extraction methods. As such, the utilization of potassium cyanide, an inhibitor of Cu/Zn SOD, and H2O2, an inhibitor of Fe and Cu/Zn SOD, in parallel triplicate wells with the existing SOD assay protocol would allow for the isolation of the activity values for each isozyme, improving the ability to understand how different like-enzymes contribute to ROS detoxification under endogenous ROS cycling (Beauchamp & Fridovich, 1971; Elstner & Heupel, 1976; Richier et al., 2003).

These results highlight avenues for continuing studies, as greater investigation into the interplay of ROS generation and detoxification during coral reproduction allude to significant inherent stress thresholds in corals. Now that these data exist, suggesting that coral in the field are undergoing cyclical variations in enzyme activity with relation to reproduction, there is a need to replicate these results in a controlled laboratory setting. This will allow us to control for environmental variables, enhancing data resolution. Although environmental factors were controlled as much as possible by experimental design (collection times, tides, and sampling around weather anomalies), additional validation of these findings in a controlled laboratory setting would be a natural progression with respect to this research. Expansion of this study through the comparison of antioxidant enzyme activity profiles over different coral species, especially those utilizing broadcast spawning, reproducing on annual scales, and those exhibiting gonochory, would allow us to understand potential interspecies variations and diversity in activity over different reproductive strategies. Furthermore, investigations of antioxidant enzyme activity on the intraspecies level in P. damicornis should be conducted across geographic boundaries to elucidate potential location-based differences in activity (Harriott, 1983; Harrison & Wallace, 1990; Stoddart & Black, 1985). Finally, though these assays provide valuable insight into coral molecular biology, additional expansion of our capacity to examine coral proteomics, such as the confirmation of protein presence and enzyme isotypes through techniques like High-Resolution Mass Spectrometry (such as with a Q-trap), would provide more fine resolution analyses of coral tissue samples and absolute mass determination. This would allow for greater flexibility to diagnose stress biomarkers through enzyme assays, independently confirmed by mass detection of enzyme proteins, that would aid in quantitatively and qualitatively evaluating enzyme variation.

Understanding the influence of reproduction on various biomarker enzymes in coral remains a poorly characterized field that merits expansion. The enzymes employed in this study have been widely applied to evaluate the effect of many abiotic stressors on coral health (Flores-Ramírez & Liñán Cabello, 2007; Higuchi et al., 2008; Liñán Cabello et al., 2010a; Liñán Cabello et al., 2010b; Richier et al., 2003; Verma, Mehta & Srivastava, 2007; Yakovleva et al., 2004). However, failing to consider reproduction as a source of inherent variation in ROS-induced stress responses presents a potential confounder for studies using these biomarkers by potential mischaracterization of natural fluctuations in antioxidant biomarkers as stress responses. Hence, consideration of reproductive cycling when surveying and comparing coral populations for variability in antioxidant enzyme activity is necessary to prevent confounding data (Agarwal, Gupta & Sharma, 2005; Agarwal, Gupta & Sikka, 2006; Fujii, Iuchi & Okada, 2005; Ramos et al., 2011; Rougée, Richmond & Collier, 2014).

Conclusions

The findings of this study illustrate significant changes in the activities of CAT, SOD, GPx, and GR in response to reproductive cycling. These data demonstrate that peaks in the activities of these enzymes correlate with reproductive peaks and troughs over monthly planulation cycles. Due to their value as bioindicators of oxidative stress, our findings demonstrate the importance of determining endogenous cycling of oxidative enzymes tied to reproduction. Such baseline data tied to homeostasis help eliminate confounding factors in studies analyzing the impact of oxidative stress on this species. These results also present greater impetus for future studies elucidating the effects of oxidative stress on reproduction and the overall health of other brooding, and perhaps broadcast spawning, coral species. Molecular tools such as those presented here provide critical data on cause-and-effect relationships between putative stressors and coral health which can be used to guide and evaluate the effectiveness of management and mitigation measures designed to protect coral reefs and those who depend on these magnificent ecosystems.

Supplemental Information

Dataset S1 Raw enzyme activity assay values

Activity values are displayed in mmols/min/mg protein for CAT and nmols/min/mg protein for GR, GPx, and SOD. Assays were rerun if replicates experienced errors, these could be due in part to issues, such as bubbling during the metabolism of substrates. If assays could not be rerun due to concerns about sample or reagent integrity, those values were excluded (−).

Click here for additional data file.

We would like to thank Narrissa P. Spies for her time and edits that significantly enhanced the quality of this manuscript. We would also like to thank Drs. Paul Jokiel and Ku‘ulei Rodgers for their assistance in facilitating sample collections and access to the sampling site. Without their support, this research would not have been possible. Due to the location of this research and cultural ties of the first author, Hawaiian ‘oli, or chants, were integrated into the collection protocol. Without a written language, Hawaiians employed ‘oli as a means of passing down knowledge in the form of orally communicated genealogies, stories, and protocols for interacting with specific daily or ceremonial practices, among other things. Prior to each collection, “E Hō Mai” and “Nā ‘Aumākua” were chanted to ask for knowledge and permission to enter the collection site, while “Oli Mahalo” was chanted following each collection to both signify the end of the sampling period and give thanks for the coral taken.

Additional Information and Declarations

Competing Interests

Author Contributions

Field Study Permissions

Data Availability

The authors declare there are no competing interests.

James W.A. Murphy conceived and designed the experiments, performed the experiments, analyzed the data, contributed reagents/materials/analysis tools, prepared figures and/or tables, authored or reviewed drafts of the paper, approved the final draft.

Abby C. Collier analyzed the data, contributed reagents/materials/analysis tools, authored or reviewed drafts of the paper, approved the final draft.

Robert H. Richmond contributed reagents/materials/analysis tools, authored or reviewed drafts of the paper, approved the final draft.

The following information was supplied relating to field study approvals (i.e., approving body and any reference numbers):

Department of Land and Natural Resources—Division of Aquatic Resources coral collection permit 2015-06 and 2015-17 (O‘ahu, HI, USA).

The following information was supplied regarding data availability:

The raw values for enzyme activity is available in Supplemental Information 1. All CAT activity is expressed in mmols/min/mg protein, while SOD, GPx, and GR activity is expressed in nmols/min/mg protein.

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
