# Peer review of "Antioxidant enzyme cycling over reproductive lunar cycles in Pocillopora damicornis"

_PeerJ, doi:10.7717/peerj.7020_

## Round 0.1 · original submission · Major Revisions

Your manuscript has been reviewed by two expert reviewers. Both reviewers stated that the study was interesting, however numerous items need to be addressed before publication. I have reviewed all of their comments as well as the manuscript and agree with both reviewers that the manuscript could be improved considerably with modifications in regards to clarity and presentation of results.

Specifically, the 11 figures should be condensed to fewer figures with multiple bar chart panels. Both reviewers stipulated that the authors provided insufficient details about the reproductive status of P. damicornis. Hence, the authors claim the observed patterns relate to reproductive cycles, but only demonstrate temporal patterns in enzymatic activity across lunar cycles.

Since the main conclusions of the study assert that changes in activities of different antioxidants are in response to reproductive cycling, it is critical to show some link between lunar cycle and reproductive status of these coral colonies. If this is not feasible with the samples you have from this study, then at the very least further discussion of this caveat needs to be addressed in the discussion.

Reviewer 1 ·

Basic reporting

On the whole the manuscript is well written. I feel there needs to be more detail provided on the reproduction of P damicornis, as it is very interesting and relevant to the results here. The monthly planulation is not universal and varies a great deal geographically.
The captions for the figures could be improved. Figures should stand alone but there is not enough detail here for that to be the case. The titles of the figures should be removed from the figure with the details in the caption. It is unusual for results to be included in the figure caption. I think the figures may look better without the gridlines.
It is unusual to place report p values in an abstract.
The authors listed all the authors for many of the citations instead of using et al. This needs to be fixed. The authors also have not used standard unit abbreviations throughout the methods section.
The aims could be explained more clearly. What are the specific questions or hypotheses for the work?

Experimental design

Authors explained the relevance of this project well and I appreciate the argument that we must understand what occurs in normal conditions prior to understanding how environmental change might affect these factors.
Methods are described well with considerable detail.

Validity of the findings

I think this is important work. The authors need to discuss whether they can be sure that their findings are actually do to the reproductive cycle of the coral rather than occurring in response to the lunar cycle. I expect that they are due to reproduction, but this design does not preclude the lunar cycle being responsible, so this possibility needs to be addressed in the discussion. It would be good to repeat this work in a location where planulation is not talking place, so that changes due to the lunar cycle can be assessed.
I think the authors need to consider the possibility that P damicornis might be spawning gametes at this location. Spawning in this species can occur in corals that are also producing brooded larvae, but it is not obvious so can easily be missed.

Additional comments

This is a good contribution to an important area where there are still considerable gaps in our understanding. I have made some editorial suggestions on the pdf.

Reviewer 2 ·

Basic reporting

In this manuscript the authors seek to characterize variability in antioxidant enzyme activity that is associated with monthly reproductive cycles in a brooding corals. Identifying the scale and timing of this variability could help investigators to separated induction of antioxidant enzymes by exogenous stressors from variation associated with endogenous cycles. In general, the language was clear, but I have detailed suggestions for improvement that will be provided at the end of the “general comments” section.
Additional references should be provided in two areas: (1) The authors have customized protocols for enzymatic activities; however, these analytical methods (e.g., choices of substrate, assay buffers, wavelengths for monitoring metabolite production or consumption) are certainly built upon previously developed methods which should be cited. (2) Related to multiple isoforms of SOD (see comment related to L464-7 in the last section).
The structure of the manuscript was appropriate. Checking for conformance to PeerJ requirements seems like it should be a job for the editorial staff.
I have several comments about figures: (1) The enzymatic activity figures are relevant (2-9, portions of 10-11); however, they are incompletely labeled. Asterisks are used to indicate statistical significance, but it should be clear from looking at the figure which specific contrasts are significant. Presently this information is only in the figure legend. A more informative presentation could be achieved by using distinct letters to indicate statistically different groups, or by marking the significant comparisons with brackets. (2) The manuscript frequently refers to reproductive and non-reproductive periods (e.g., Figure 4 caption), but these are not clearly defined or labeled on the figures. Are only the first quarter points considered reproductive, or (as suggested by Figure 4 caption) is new moon also included in the “reproductive” period? (3) I do not consider the western blots within figures 10 and 11 to be relevant. My specific concerns about the western blots will be included in comments on experimental design. In addition, the western blots are not described in the figure legends. (2) All the plots of enzymatic activity are necessary, but they could be more easily interpreted if they were combined into a smaller number of figures that each contain multiple sub-plots. One possibility would be to group Figures 2, 4, 6, 8 and 10 into one figure (because a reader would want to look at difference among enzymes in monthly timing) and Figures 3,5,7,9 and 11 into a second figure (for similar reasons).
Raw data are provided. I suggest that the Excel workbook containing raw enzyme values should be rearranged such that it is clear which values correspond to which colonies. This might be easiest if the columns corresponded to colonies and the rows corresponded to individual measurements (or the reverse). This is important given that the experimental design employs repeated sampling of the same colonies over time (as discussed below).

Experimental design

The manuscript contains original primary research that is within the scope of PeerJ (this also seems like it should be a job for the editor). The research question is well-defined and meaningful. While the rationale was clear within the main document, it was less clear within the abstract. A brief justification should be provided to indicate that there is reason to think antioxidant enzyme activity might vary over reproductive cycles. I do have some specific concerns about the methods and they ways that they are presented.
Individual colonies were sampled repeatedly over time, but the data were analyzed using one-way ANOVA. I believe that a repeated measures analysis is necessary given the experimental design used.
The methods used for waveform analysis are unclear. A frequency of 1 is equivalent to a perfect sinusoidal fit, but it is unclear how this parameter scales (i.e., 1.72 is considered a “not strong” fit in L353, but it is unclear what basis is used to designate a curve fit as strong or weak).
I recognize the potential value of western blot assays, the large amount of work the investigators devoted to this aspect of the project, and the frustration they likely felt when faced with variation in antibody batches. Still, I do not feel that the Western blot analysis makes a meaningful contribution to the manuscript. I have several concerns: (1) It is not clear what criteria the authors used in their preliminary assays to determine that “all antibodies…work in this coral species” [L271]. (2) It was later found that the authors could not reliably measure three of the four targeted enzymes, so no data are presented. There is no need to discuss these measurements that ultimately didn’t work. As is the description and discussion of failed measurements take up substantial space and result in substantial repetition within the discussion. I suggest that the failed assays not be mentioned in the methods or results. In addition, I suggest that these failed efforts will be briefly discussed (all three discussed together in one brief section). (3) The apparently even levels of SOD protein do not provide any additional insight and the data are relatively limited. No loading control is shown and there is no replication. How was the expected size of the protein calculated, or was this inferred from previous studies? There are multiple forms of SOD in most cnidarians (as well as their symbionts), so I wouldn’t necessarily expect only a single band. (4) I might have missed this but I’m not sure that the supplementary figure showing the full blot is referenced within the paper (or perhaps in the figure legend). If these data are retained, it’s important for the reader to know that they can look at the full blot.

Validity of the findings

Some of my concerns are reflected in the preceding section. I recommend that the authors incorporate repeated measures into their statistical analysis. Additional information is needed to enable full evaluation of the waveform analysis. Additional information would also be needed to ensure that the SOD western blot analyses are valid (although I have recommended removing these).
Temporal patterns in enzymatic activity are interpreted in the context of lunar reproductive cycles. There is no direct demonstration of causation. The interpretation is strengthened by the repeated sampling through two lunar cycles and the waveform analysis. No environmental parameters are measured, and the reproductive status of each colony is unknown. The authors do address some of this uncertainty late in the discussion (L478-481). Some qualification of the interpretations should be given earlier in the manuscript (perhaps around L400). As a suggestion, the authors point toward controlled laboratory studies as a natural future extension of the work. The authors might consider also adding monitoring of reproductive output. This would be particularly powerful within the context of a repeated measures design, such as the authors have used.

Additional comments

My most substantial comments have been incorporated into the preceding sections. I think data are valuable, but would particularly like to see revised statistical analyses and presentation of significance results on figures.
L54 language imprecise. Suggest something like “wide application in assessing responses to diverse stressors” or “their involvement in physiological responses to a variety of stressors”
L74 This is intended as a question and not a criticism. Regarding “prior to adopting these enzymes into our suite of diagnostic tools.” Arguably the authors have already done this to some extent (Downs et al. 2006 & 2012 and Rougee et al 2006 and 2012 include Richmond as a co-author). I have not cross-checked all of these references in detail, but at least some of them include measurements of some antioxidant enzymes (protein levels rather than activity, I think) in the context of ecotoxicology studies. The authors can make their own decision about this, but it might be useful to edit the statement. Possibly something like “Prior to continuing and expanding our use of these enzymes…” (?)
L78 This use of “defense enzyme activity” is vague. Suggest “cyclical variation in the activity of xenobiotic metabolizing enzymes…”
L88 I don’t think ROS has been defined yet (defined in L92).
L96-7 I don’t know what is meant by “systems other than those found in Cnidarians” (what “systems”?) I think the authors mean, “studies conducted in non-cnidarian species” or similar.
L111 Presumably antioxidant enzymes exist for many purposes, some of which are in included in this sentence.
L117 enzymatic activities are not necessarily “enzymatic stress levels”
L118 I’m not sure what is meant by “application”. Authors might simply mean “distribution” or they might mean both a broad distribution, and the frequency with which P. damicornis has been used in physiological or toxicological studies.
L119 As written, it might imply to a non-specialist that brooding is unique to P. damicornis. I suggest “unlike many other common reef-building corals…”
L125-126 I have no idea what the authors are referring to here. If previous studies have demonstrated monthly cycles in antioxidant enzymes, they should be cited and clearly described here and differentiated from the current study. Similarly, which studies have looked for and failed to find an annual cycle in antioxidant enzymes? Such a cycle might be expected in response to seasonal changes in light and temperature (as indicated by the authors in other parts of the manuscript). L473-474 suggest that monthly and/or annual cycles have already been characterized for at least some of the antioxidant enzymes. The manuscript needs to clearly indicate how the present study distinct from the work by Rougee et al. 2014.
L128 syntax doesn’t make sense here.
L131 I don’t understand this either. Internal cycling is a natural phenomenon. What stresses are the authors talking about? Presumably they are actually interested in anthropogenic stressors as well.
L143-145 I didn’t fully understand this. The authors discuss variation along the length of the branch, but it’s not clear which portion of the branch was used (presumably a fragment that incorporated both the tip and mid-branch polyps)? I don’t really know how the sampling design reduced variation. Samping branches from different parts of the colony is different from sampling tissue from different portions of the branch. (I know the authors know this, it just needs to be explained more clearly).
L139-153 In the comments to reviewers, the authors included a justification for the sample size based on preliminary power analyses and other considerations. This information could be included somewhere in this paragraph.
L154-6 Two very different ideas are lumped into this sentence. Please separate. The first part probably belongs in the previous paragraph.
L157-8 It’s stated twice that the samples were stored at -80. I think this could be streamlined (I think the authors mean that they were frozen immediately upon collection and then later after pulverization, but the description could still be simplified). I don’t think naming the freezer model is necessary.
L162-3 This largely repeats L157-8 and the two descriptions of pulverization should be combined.
L172 It is not necessary to describe the preservation of the zoox samples because they were not used in any analyses within this manuscript.
L178-9 I’m pretty sure this also includes buffers
L227 microtiter
243 substrate [comma] and buffer
L245 Since preceding sentence is plural and two classes of GPx are measured, I suggest “GPx enzymes use…” or “Both forms of GPx use…”
249 decreases are proportional
L250 96-well microtiter
L256, L261 the NaN and CHP (and maybe some other components) are only added to assays for one of the GPx forms
L266 I have no idea what is meant by “Materials…were sourced in-house” I think the sentence could just be deleted. (but I also think this section should be deleted).
L266-310 I’m not commenting in detail because I feel this section should be deleted.
L316 I don’t think the variance itself is significant. I think the differences in the means are significant in the context of the variance associated with each group (but as previously mentioned, I think a different analysis should be conducted).
L317-320 Were these analyses really conducted in GraphPad Prism as indicated in L313? I could not find a waveform analysis function, although it might be possible do perform one.
L320 I don’t think it’s necessarily true that values above or below these are caused by external factors, just as it’s not necessarily true that all this variation was caused by “internal” factors.
L338 suggest replacing “cycles” with “phases”
L358 suggest “across all moon phases and throughout the acute collections”
L361 Conversely to what? Several patterns have been described.
L363 please be more specific than “collection periods throughout the sampling cycle” Significantly higher than all other sampling times?
L395 is “potential” needed here?
L397 changes in activity of antioxidant enxymes is not necessarily “stress flux”
L399 suggest more precise wording than “reproductive baselines”
L402 write out species
L407 Not sure what is meant by “greater” More importantly?
L420-422 This is not necessarily true. For example, an increase to 66 during a “low” part of the cycle might be due to external stimuli. Reproduction varies on a lunar cycle, but also exhibits some annual periodicity. In the winter a smaller change might indicate a response to some external stimulus.
L422-425 This statement is relevant to all enzymes and not specific to CAT. Suggest it be moved to an introductory or concluding paragraph.
L444-6 Not sure exactly what is meant here. Are the authors suggesting that with some additional experimentation, wave fits would be useful, or that some other curve fitting could be done? If additional experiments are needed, how should they differ from what has already been done?
L455-457 Needs some edits. As is current syntax is confusing “Previous studies analyzing GPx activity…confirmed activity and presence”
L460 agreement problem: increases…that (plural/singular)
L463 not sure what is meant but “adaptive function of their cellular detoxification”
L464-467 Authors should cite some work from Allemand lab in discussion of multiple SOD isoforms. Although the work was conducted in anemones, it is in my mind the most thorough study of SOD activity conducted to date in cnidarians. I don’t have access to full internet right now, but I think the most appropriate reference is likely Richier et al. 2003 (which the authors have cited elsewhere).
L473-4 Within this sentence, it is unclear how the present study is different from what was already done by Rougee et al. (see similar comment L125)
L478-481 To do this, changes in environmental conditions should also be measured.
L489-490 I don’t agree that the blots confirm that the assays were effectively designed.
L500-501 The limitations of the western blots don’t necessarily impede proteomic assays. The probably are specific to antibody-based measurements. Proteomics more commonly refers to mass spec-based methods (such as those discussed in L502-3).
Figure 1 caption: replaced “of interest in “ with “sampled during”
Fig 2 caption (and perhaps other captions) “moon phase cycle” doesn’t make much sense. Suggest “moon phase” or “phase of the lunar cycle”

---

## Round 0.2 · Minor Revisions

Both reviewers highlight significant improvements in the revised manuscript. Before publication, please resolve the comments listed by Reviewer 2. In particular, please ensure figures 2 and 3 are correct and consistent with regard to the text in results and discussion sections. Also, the asterisks denoting significance are awkward in figure 2. I recommend using letter codes to identify which time points differ significantly from others. Also, be sure to correct the text as outlined by reviewer 2.
Additionally, please check that all citations in the manuscript are listed in the Reference section. I noticed several that were not present (e.g., Aebi 1984; Lawrence & Burk 1976; etc).

Please include a point by point rebuttal with your revision. Nice work and I look forward to seeing the finalized article!

Reviewer 1 ·

Basic reporting

I am satisfied with the changes made by the authors and feel that the manuscript is much clearer in its intent, methods and results now.

Experimental design

I have no further comments. I am happy with the changes made.

Validity of the findings

The results are now more clearly presented and discussed than they were in the first version of the manuscript. I have no further comments.

Additional comments

The authors have made a thorough attempt to consider all comments made by both reviewers in their new version of this manuscript. I am pleased with this new version and feel that it is ready for publication. I have no further suggestions or criticisms of the manuscript.

Reviewer 2 ·

Basic reporting

No comment

Experimental design

I understand from their response that the authors were not able to conduct a repeated measures analysis due to analytical issues. This should be mentioned somewhere in the statistical section. It is not strictly correct to conduct one-way ANOVAs because independence is an assumption of the test.

Validity of the findings

No comment

Additional comments

The manuscript is much improved, but there are issues that remain to be addressed.
1. I think there is at least one error in Figure 2. To really understand what is going on, it would be helpful (1) if authors referred explicitly to subfigures in text, and (2) if subfigures were arranged in the order that they are discussed. As is, SOD is 2b but is discussed last. The text does not match what is shown in the figure.
2. The same is true for Figure 3. L318 says SOD activity is higher on day 5 than days 3 and 4. That is not what is shown in Figure 3b (day 5 is the lowest in the figure.
3. Regarding waveform analysis, it is helpful that the authors explained to me in the comments how they did the analysis. This information needs to be included in the text such a reader could reproduce it.
L16 [,] and action
L68 suggest replacing “expression” with “activity” (since activity was measured in this study)
L126 suggest authors consider a different phrase than “inherent stress.” This is just my opinion. I would suggest a phrase like “inherent variation in redox state” or “inherent variation in cellular physiology” or “endogenous cycles in cellular physiology”
L223 suggest removing “Using the following protocol”
L394-408 Repeating a variation on my comments on the previous draft. I think it is very important to mention that SODs are a family of enzymes that even vary in their subcellular localization. It’s not particularly surprising that there could be cycles driven by multiple factors on multiple time scales. This is true of some of the other enzymes measured too. It should be clear to readers.
L424-430 The idea that proteomics (given sufficient analytical ability) would be useful is good, but the transition doesn’t make sense as written. I wonder if this text is left over from earlier version that included the westerns. There is nothing wrong with the overall text, it just needs to flow more smoothly. “This work” doesn’t particularly highlight deficiencies in capacity to do proteomics. It’s more that the proteomics would be a great follow-up and the abilities currently aren’t there.
L449 agreement problem. Data…help
Figures 2 and 3 Why do the captions say “CI = 95%” if confidence intervals aren’t shown? According to the caption, the error bars are standard deviation.

---

## Round 0.3 · accepted · Accept

Thank you for your careful revisions. This manuscript will provide a great contribution to PeerJ! I really liked your study and am considering trying some of the biochemical assays to see if any correlate with these spikes of dissolved oxygen we observe on reefs at night.

Best wishes,
Linda

#